# Theory of Functional Connections Applied to Linear ODEs Subject to Integral Constraints and Linear Ordinary Integro-Differential Equations

**Mario De Florio** [1] , **Enrico Schiassi** [1] , **Andrea D'Ambrosio** [2] , **Daniele Mortari** [3,*] **and Roberto Furfaro** [1]

1   Department of Systems and Industrial Engineering, The University of Arizona, 1127 E. James E. Rogers Way, Tucson, AZ 85721, USA; mariodf@email.arizona.edu (M.D.F.); eschiassi@email.arizona.edu (E.S.); robertof@email.arizona.edu (R.F.)
2   School of Aerospace Engineering, Università degli Studi di Roma "La Sapienza", Via Salaria 851, 00138 Rome, Italy; andrea.dambrosio@uniroma1.it
3   Department of Aerospace Engineering, College of Engineering, Texas A&M University, 401 Joe Routt Blvd., College Station, TX 77843, USA
*   Correspondence: mortari@tamu.edu

**Abstract:** This study shows how the Theory of Functional Connections (TFC) allows us to obtain fast and highly accurate solutions to linear ODEs involving integrals. Integrals can be constraints and/or terms of the differential equations (e.g., ordinary integro-differential equations). This study first summarizes TFC, a mathematical procedure to obtain *constrained expressions*. These are functionals representing *all functions satisfying a set of linear constraints*. These functionals contain a free function, $g(x)$, representing the unknown function to optimize. Two numerical approaches are shown to numerically estimate $g(x)$. The first models $g(x)$ as a linear combination of a set of basis functions, such as Chebyshev or Legendre orthogonal polynomials, while the second models $g(x)$ as a neural network. Meaningful problems are provided. In all numerical problems, the proposed method produces very fast and accurate solutions.

**Keywords:** Theory of Functional Connections; Ordinary Differential Equations; integro-differential equations; Extreme Learning Machine; numerical methods

## 1. Introduction

This paper shows how to solve linear Ordinary Differential Equations (ODEs) and linear Integro-Differential Equations (IDEs) using a new mathematical framework to perform functional interpolation, called *Theory of Functional Connections* (TFC). TFC derives functionals, called *constrained expressions*, containing a free function and representing all possible functions satisfying a set of linear constraints [1–4]. The most important feature of the constrained expressions is: *they always satisfy all the constraints no matter what the free function is*.

Although it was recently developed (2017), TFC has already found several applications, especially in solving differential equations [5–8]. The free function can be expressed by any set of linearly independent functions, such as an expansion of orthogonal polynomials (e.g., Chebyshev, Legendre, etc.) or Neural Networks (NN), such as shallow NN with random features, or Deep NNs (DNNs). When the free function is expanded by a set of orthogonal polynomials the method is here identified as "classic TFC". When shallow NNs with random features are used, the method has been identified as Extreme-TFC (X-TFC) [9], and when DNNs are used the method is called Deep-TFC [10].

The best expansion of the free function depends on the differential equation considered. Usually, for linear/nonlinear ODEs and simple bivariate PDEs, orthogonal polynomials represent the best choice in terms of accuracy. Indeed, orthogonal polynomials are generally the best mathematical tools for approximating and for convergence properties [11,12].

Nevertheless, for multivariate and more complex PDEs, orthogonal polynomials suffer the "curse of dimensionality". For these problems, NNs represent a better choice as they better tolerate the curse of dimensionality.

In all previous applications, the free function has been expressed as a linear combination of known basis functions and unknown constant coefficients. These coefficients are then estimated by least-squares for linear Differential Equations (DEs) [5] and by nonlinear least-squares for nonlinear DEs [6]. The problems considered in this work will be solved with classic TFC using Chebyshev orthogonal polynomials and X-TFC.

Note that X-TFC and Deep-TFC are also identified as Physics-Informed Neural Networks (PINN) frameworks [9]. PINNs are recently developed machine-learning methods that employ NNs for data-physics-driven regression, data-physics-driven solution of DEs, and data-physics-driven discovery of parameters governing DEs [13]. Since X-TFC and Deep-TFC use NNs as free function, they are considered part of the PINNs family. Thanks to the constrained expression, developed by TFC, both X-TFC and Deep-TFC are more robust and accurate than the standard PINN frameworks as introduced by Raissi et al. [13].

TFC has been developed for univariate and multivariate scenarios [2,7,8] to solve a variety of mathematical problems: a homotopy continuation algorithm for dynamics and control problems [14], domain mapping [15], data-driven parameters discovery applied to epidemiological compartmental models [16], transport theory problems such as radiative transfer [17] and rarefied-gas dynamics [18], nonlinear programming under equality constraints [19], Timoshenko–Ehrenfest beam [20], boundary-value problems in hybrid systems [21], eighth-order boundary value problems [22], and in Support Vector Machine [23]. TFC has been widely used for solving optimal control problems for space application, solved via indirect methods [24]: orbit transfer and propagation [25–28], energy-optimal in relative motion [29], energy-optimal and fuel-efficient landing on small and large planetary bodies [30,31], the minimum time-energy optimal intercept problem [32].

The aim of this paper is to show how TFC can accommodate integral constraints in linear differential equations and solve linear integro-differential equations, is organized as follows. In Section 2, a summary of the TFC framework is provided, with the explanation of how to derive constrained expressions. In Section 3, the application of TFC to solve ODEs with integral constraints is shown, and some case problems are reported as examples. In Section 4, the TFC framework is used to solve linear integro-differential equations.

In all test problems considered in this article, the results have been obtained using MATLAB R2020a software on an Intel Core i7-9700 CPU PC with 64 GB of RAM. The results' accuracy is provided in terms of the absolute error. That is,

$$err = |y_{\text{TFC}} - y_{\text{true}}|,$$

where $y_{\text{TFC}}$ is the TFC approximated solution, and $y_{\text{true}}$ is the true analytical solution.

Since the classic TFC uses Chebyshev or Legendre orthogonal polynomials as a basis set, the final Appendix A provides a definition, orthogonality, derivatives, and integral expressions and properties for both Chebyshev and Legendre orthogonal polynomials.

## 2. Theory of Functional Connections Summary

A mathematical generalization of interpolation, called *Theory of Functional Connections*, has recently been developed and successfully applied to solve, by least-squares, initial, boundary, and multi-value problems of linear [5] and nonlinear [6] ODEs and PDEs [8,10]. The theory has been developed for univariate [1] and multivariate rectangular domains [2,8].

The generalization of interpolation consists of a mathematical procedure to obtain analytical expressions describing *all* possible functions subject to *n* linear constraints. These expressions are functionals that are called *constrained expressions*. Two formally equivalent approaches [1,8] can be used to derive them. These are,

$$y(x, g(x)) = g(x) + \sum_{k=1}^{n} \eta_k(x, g(x)) \, s_k(x), \tag{1}$$

$$y(x, g(x)) = g(x) + \sum_{k=1}^{n} \phi_k(x, \mathbf{s}(x)) \, \rho_k(x, g(x)), \tag{2}$$

where $n$ is the number of the linear constraints, $g(x)$ is a function that can be freely chosen, $\eta_k(x, g(x))$ are *functional coefficients*, $\mathbf{s}(x) = \{s_1(x), \cdots, s_n(x)\}$ is a set of $n$ user-defined *support functions* that must be linearly independent (necessary conditions), $\phi_k$ are *switching functions*, and $\rho_k(x, g(x))$ are *projection functionals*.

By imposing the $n$ constraints to Equation (1), the values of the $n$ functional coefficients, $\eta_k(x, g(x))$, are obtained for some set of $n$ user-assigned functions, $s_k(x)$. Once the $n$ functional coefficients, $\eta_k(x, g(x))$, are computed, by substituting them back into Equation (1) we obtain the constrained expression. To give an example, using $s_1(x) = x$ and $s_2(x) = x^2$, the constrained expression,

$$f(x, g(x)) = g(x) + \underbrace{\frac{x\,(2x_2 - x)}{2(x_2 - x_1)}}_{\phi_1(x)} \underbrace{(f_{x1} - g_x(x_1))}_{\rho_1(x, g(x))} + \underbrace{\frac{x\,(x - 2x_1)}{2(x_2 - x_1)}}_{\phi_2(x)} \underbrace{(f_{x2} - g_x(x_2))}_{\rho_2(x, g(x))},$$

where the switching functions, $\phi_k(x)$, and projection functionals, $\rho_k(x, g(x))$, are identified, always satisfies the two derivative constraints, $f_x(x_1) = f_{x1}$ and $f_x(x_2) = f_{x2}$, no matter what $g(x)$ is. In this article, the use of both formulations, provided by Equations (1) and (2), will be shown.

The projection functional, $\rho_k(x, g(x))$, projects the free function, $g(x)$, on the $k$-th constraint. Here is an example of projection functionals associated to a set of constraints,

$$\begin{cases} f(3) = 9 \\ 3f(0) - f(2) = 0 \\ \ddot{f}(1) = -1 \end{cases} \qquad \rightarrow \qquad \begin{cases} \rho_1 = 9 - g(3) \\ \rho_2 = -3g(0) + g(2) \\ \rho_3 = -1 - \ddot{g}(1) \end{cases} .$$

As for the switching functions, $\phi_k(x)$, they are expressed as a linear combination of a set of support functions, $\mathbf{s}(x)$. These functions satisfy $\phi_k(x) = 1$ when the $k$-th constraint is verified and $\phi_k(x) = 0$ when any other constraint is verified. For instance, given the support functions,

$$s_1(x) = x, \qquad s_2(x) = e^x, \qquad \text{and} \qquad s_3(x) = \sin x,$$

the switching functions are computed as $\phi_j(x) = \sum_{i=1}^{3} \alpha_{ji} \, s_i(x)$, where the $\alpha_{ji}$ coefficients are computed by inverting the support matrix. For this example,

$$\begin{bmatrix} \alpha_{11} & \alpha_{12} & \alpha_{13} \\ \alpha_{21} & \alpha_{22} & \alpha_{23} \\ \alpha_{31} & \alpha_{32} & \alpha_{33} \end{bmatrix} \begin{bmatrix} s_1(3) & 3s_1(0) - s_1(2) & \ddot{s}_1(1) \\ s_2(3) & 3s_2(0) - s_2(2) & \ddot{s}_2(1) \\ s_3(3) & 3s_3(0) - s_3(2) & \ddot{s}_3(1) \end{bmatrix} = \begin{bmatrix} 1 & 0 & 0 \\ 0 & 1 & 0 \\ 0 & 0 & 1 \end{bmatrix}.$$

Reference [4] presents, in detail, how to derive the TFC constrained expressions using the formalism defined by Equation (2).

Constrained expressions have been used to solve differential equations. This is done by expressing the solution using the TFC constrained expressions from Equation (1) or, equivalently, from Equation (2). These functionals allow us to reduce the whole functions space to only the functions subspace satisfying the constraints. This is particularly important when solving differential equations. In fact, when substituting these functionals into the DE, a new differential equation is obtained in terms of $g(x)$. This new DE is subject to *no constraints* because the constrained expression fully satisfies the constraints. The unknown

free function, $g(x)$, is then expressed as a linear combination of basis functions. In the classic TFC, $g(x)$ is a linear combination of orthogonal polynomials. That is,

$$g(x, \boldsymbol{\xi}) = \sum_{j=0}^{m} \xi_j \, h_j(x) = \boldsymbol{\xi}^{\mathrm{T}} \boldsymbol{h}(x),$$

where $m$ is the number of the basis functions $h_j(x)$ (orthogonal polynomials). In X-TFC, $g(x, \boldsymbol{\xi})$ is a shallow NN trained with Extreme Learning Machine (ELM) [33],

$$g(x, \boldsymbol{\xi}) = \sum_{j=1}^{L} \xi_j \, \sigma_j(w_j \, x + b_j) = \boldsymbol{\xi}^{\mathrm{T}} \boldsymbol{\sigma}(x) = \boldsymbol{\xi}^{\mathrm{T}} \left\{ \begin{array}{c} \sigma_1(x) \\ \vdots \\ \sigma_L(x), \end{array} \right\}$$

where $L$ is the number of hidden neurons, $w_j \in \mathbb{R}$ is the input weights vector connecting the $j$-th hidden neuron and the input nodes, $\xi_j \in \mathbb{R}$ with $j = 1, \cdots, L$ is the $j$-th output weight connecting the $j$-th hidden neuron and the output node, and $b_j$ is the bias of the $j$-th hidden neuron, $\sigma_j(\cdot)$ are activation functions, and $\boldsymbol{\sigma} = \{\sigma_1, \cdots, \sigma_L\}^{\mathrm{T}}$. According to the ELM algorithm [33], biases and input weights are randomly selected and not tuned during the training, thus they are known hyper-parameters. The activation functions, $\sigma_j(\cdot)$, are also known as they are user selected. Thus, the only unknown NN hyper-parameters to compute are the output weights $\boldsymbol{\xi} = \{\xi_1, \cdots, \xi_L\}^{\mathrm{T}}$. Therefore, for both classic TFC and X-TFC, the unknown vector, $\boldsymbol{\xi}$, which actually constitutes the *only* unknown of our problem, is then estimated numerically, as for instance by least-squares for linear [5] and nonlinear [6] differential equation s.

In general, the independent variable of the DE (for instance, time) is defined in the $t \in [t_0, t_f]$ range, while the selected basis functions may be defined as a different range, $x \in [x_0, x_f]$ (for instance, Chebyshev and Legendre orthogonal polynomials are defined in the $x \in [-1, +1]$). Thus, a change of variable is needed. The most simple mapping between these two variables is linear,

$$x = x_0 + \frac{x_f - x_0}{t_f - t_0}(t - t_0) \qquad \longleftrightarrow \qquad t = t_0 + \frac{t_f - t_0}{x_f - x_0}(x - x_0). \tag{3}$$

By setting the range ratio, $c = \dfrac{x_f - x_0}{t_f - t_0}$, the derivatives in terms of the new variable are

$$\frac{\mathrm{d}^k f}{\mathrm{d}t^k} = c^k \, \frac{\mathrm{d}^k f}{\mathrm{d}x^k}, \tag{4}$$

and the derivative constraints can be written as:

$$\left. \frac{\mathrm{d}^k f}{\mathrm{d}t^k} \right|_{t_i} = f_{t_i}^{(k)} = c^k \left. \frac{\mathrm{d}^k f}{\mathrm{d}x^k} \right|_{x_i} = c^k \, f_{x_i}^{(k)}. \tag{5}$$

The change of variable in integrals takes advantage from the fact that the mean value is independent from the independent variable. Therefore,

$$\frac{1}{t_f - t_0} \int_{t_0}^{t_f} f(t) \, \mathrm{d}t = \frac{1}{x_f - x_0} \int_{x_0}^{x_f} f(x) \, \mathrm{d}x \qquad \rightarrow \qquad c \int_{t_0}^{t_f} f(t) \, \mathrm{d}t = \int_{x_0}^{x_f} f(x) \, \mathrm{d}x.$$

When expressing the free function as $g(x, \boldsymbol{\xi}) = \boldsymbol{\xi}^{\mathrm{T}} \boldsymbol{h}(x)$, then, the derivatives of $g(x, \boldsymbol{\xi})$ can be written as:

$$\frac{\mathrm{d}^k g}{\mathrm{d}x^k} = \boldsymbol{\xi}^{\mathrm{T}} \frac{\mathrm{d}^k \boldsymbol{h}}{\mathrm{d}x^k} \qquad \text{and} \qquad \left. \frac{\mathrm{d}^k g}{\mathrm{d}x^k} \right|_{x_i} = \boldsymbol{\xi}^{\mathrm{T}} \left. \frac{\mathrm{d}^k \boldsymbol{h}}{\mathrm{d}x^k} \right|_{x_i}.$$

This procedure can be applied to linear differential equations with non-constant coefficients. The final expression obtained is linear in terms of the unknown vector, $\boldsymbol{\xi}$, and can be written as:

$$\boldsymbol{a}^{\mathrm{T}}(x)\,\boldsymbol{\xi} = b(x). \tag{6}$$

To solve the problem numerically, this new equation is discretized for a set of $N$ distinct values of $x$. A different discretization scheme can be used. Usually, the discretization points are either randomly selected or linearly uniformly spaced. When using Chebyshev polynomials the best discretization is to use the zeros of the Chebyshev polynomials, also called Chebyshev points or nodes or, more formally, Chebyshev–Gauss points [12]. They are defined by the cosine distribution,

$$x_k = \cos\left(\frac{(2k-1)\pi}{2N}\right) \qquad \text{for} \qquad k = 1, \cdots, N.$$

By specifying Equation (6) for these $x_k$ values a system of $N$ linear equations is obtained in $m$ (or $L$) unknowns that is then solved for $\boldsymbol{\xi}$ by least-squares. Several least-squares methods can be used to solve (6). The optimal least-squares method to use for each problem depends on the problem itself and on what TFC technique is used to solve the problem (e.g., in this paper, classic TFC or X-TFC). For instance, for the classic TFC and linear DEs, our analysis identifies the QR decomposition on a scaled coefficient matrix as the best approach minimizing the condition number of the matrix to invert.

## 3. TFC for ODEs with Integral Constraints

In this section, we show how TFC is applied to solve linear ODEs with integral constraints. After showing how to derive the constrained expression when dealing with a general integral constraint, we will solve a couple of linear ODEs subjects to boundary conditions and integral constraints.

For the first two examples in this section, we will derive the constrained expression by following the formulation of Equation (1). For subsequent examples and problems, however, the second formulation, Equation (2), will be adopted.

### 3.1. Definite Integral Constraint

Let us consider the integral constraint,

$$\int_a^b f(x)\,\mathrm{d}x = I. \tag{7}$$

The constrained expression has the form,

$$f(x, g(x)) = g(x) + \eta_1\,s_1(x), \tag{8}$$

where $g(x)$ is a free function and $s_1(x)$ is a user-defined support function, and $\eta_1$ is a coefficient that is derived by imposing the constraint. By integrating Equation (8) the following equation:

$$I = \int_a^b g(x)\,\mathrm{d}x + \eta_1 \int_a^b s_1(x)\,\mathrm{d}x$$

is obtained, which can be rearranged to obtain the expression for $\eta_1$,

$$\eta_1 = \frac{I - \int_a^b g(x)\,\mathrm{d}x}{\int_a^b s_1(x)\,\mathrm{d}x}.$$

Substituting this value into Equation (8), we obtain a constrained expression for $f(x)$, that is, an expression that always satisfies the integral constraint for any expression of $g(x)$,

$$f(x, g(x)) = g(x) + \frac{s_1(x)}{\int_a^b s_1(x) \, \mathrm{d}x} \left( I - \int_a^b g(x) \, \mathrm{d}x \right). \tag{9}$$

This function becomes undefined when $\int_a^b s_1(x) \, \mathrm{d}x = 0$, and thus $s_1(x)$ must be selected to avoid this condition. By simply selecting $s_1(x) = 1$ Equation (9) becomes,

$$f(x, g(x)) = g(x) + \frac{1}{b - a} \left( I - \int_a^b g(x) \, \mathrm{d}x \right).$$

### 3.2. Integral and Linear Constraints

In this second example, let us consider the more complex case where, in addition to the integral constraint given in Equation (7), we also consider the additional linear constraints,

$$\alpha f(x_0) + \beta f(x_f) = 1,$$

where $\alpha$, $\beta$, $x_0$, and $x_f$ are all assigned. A sketch of this example is shown in Figure 1.

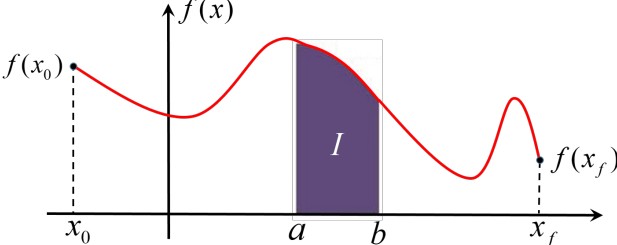

**Figure 1.** Integral and linear constraints example.

The constrained expression has the form,

$$f(x, g(x)) = g(x) + \eta_1 \, s_1(x) + \eta_2 \, s_2(x). \tag{10}$$

Then, by applying the constraints the system of linear equations,

$$\left\{ \begin{array}{l} 1 - \alpha \, g_0 - \beta \, g_f \\ I - \int_a^b g(x) \, \mathrm{d}x \end{array} \right\} = \left[ \begin{array}{cc} \alpha \, s_1(x_0) + \beta \, s_1(x_f) & \alpha \, s_2(x_0) + \beta \, s_2(x_f) \\ \int_a^b s_1(x) \, \mathrm{d}x & \int_a^b s_2(x) \, \mathrm{d}x \end{array} \right] \left\{ \begin{array}{l} \eta_1 \\ \eta_2 \end{array} \right\} \tag{11}$$

is obtained. This system tells us that $s_1(x)$ and $s_2(x)$ functions can be any functions with the exception of those making the matrix singular. The matrix singularity occurs when

$$\left( \alpha \, s_1(x_0) + \beta \, s_1(x_f) \right) \int_a^b s_2(x) \, \mathrm{d}x = \left( \alpha \, s_2(x_0) + \beta \, s_2(x_f) \right) \int_a^b s_1(x) \, \mathrm{d}x.$$

For instance, by selecting $s_1(x) = 1$ and $s_2(x) = x$, the previous condition becomes,

$$(\alpha + \beta)(b + a) = 2(\alpha \, x_0 + \beta \, x_f)$$

which imply that, in order to avoid singularity, the following relationship must be satisfied,

$$\{\alpha, \quad \beta\} \left\{ \begin{array}{l} b + a - 2\alpha x_0 \\ b + a - 2\beta x_f \end{array} \right\} \neq 0.$$

When this condition is satisfied, the matrix of Equation (11) can be inverted and the coefficients, $\eta_1$ and $\eta$, can be computed. Then, the constrained expression for this problem is obtained by substituting the expressions found for $\eta_1$ and $\eta_2$ in Equation (10).

Problem #1

Now, let us consider the following DE to be solved on the $t \in [0, \pi]$ range, with a constraint in the form of an integral over the integration range,

$$\ddot{f} + f = 0 \quad \text{subject to:} \quad \begin{cases} f(0) = 1 \\ \displaystyle\int_0^\pi f(t)\, \mathrm{d}t = \pi \end{cases},$$

whose analytical solution is

$$f = \frac{\pi}{2}\sin t + \cos t.$$

In this problem, we build the constrained expression according to the formulation of Equation (2), where the *projection functionals* $\rho_k(x, g(x))$ are given by

$$\begin{cases} \rho_1 = 1 - g_0 \\ \rho_2 = \pi - \displaystyle\int_0^\pi g(t)\, \mathrm{d}t \end{cases}.$$

Given the support functions,

$$s_1(t) = 1, \quad \text{and} \quad s_2(t) = t,$$

the switching functions are computed as $\phi_j(t) = \displaystyle\sum_{i=1}^{2} \alpha_{ji} s_i(t)$, where the $\alpha_{ji}$ coefficients are computed just by inverting a matrix. For this problem,

$$\begin{bmatrix} \alpha_{11} & \alpha_{12} \\ \alpha_{21} & \alpha_{22} \end{bmatrix} \begin{bmatrix} s_1(0) & \displaystyle\int_0^\pi s_1(\tau)\, \mathrm{d}\tau \\ s_2(0) & \displaystyle\int_0^\pi s_2(\tau)\, \mathrm{d}\tau \end{bmatrix} = \begin{bmatrix} 1 & 0 \\ 0 & 1 \end{bmatrix}.$$

Then, the constrained expression can be built as:

$$f(t, g(t)) = g(t) + \phi_1(t)(1 - g_0) + \phi_2(t)\left(\pi - \int_0^{+\pi} g(\tau)\, \mathrm{d}\tau\right),$$

where the switching functions are

$$\phi_1(t) = \frac{\pi - 2t}{\pi} \quad \text{and} \quad \phi_2(t) = \frac{2t}{\pi^2}.$$

Now, expressing the free function according to $g(t, \boldsymbol{\xi}) = \boldsymbol{\xi}^{\mathrm{T}} \boldsymbol{h}(x(t))$ the terms of the DE become:

$$f(x, \boldsymbol{\xi}) = \left\{ \boldsymbol{h}(x) - \phi_1(x)\boldsymbol{h}_0 - \phi_2(x)\frac{1}{c}\int_{-1}^{1} \boldsymbol{h}(x)\, \mathrm{d}x \right\}^{\mathrm{T}} \boldsymbol{\xi} + \phi_1(x) + \pi\phi_2(x)$$

$$\ddot{f}(x, \boldsymbol{\xi}) = c^2\, \ddot{\boldsymbol{h}}(x)^{\mathrm{T}} \boldsymbol{\xi},$$

where the boundary conditions have been mapped to $x \in [x_0, x_f]$. For classic TFC $x \in [-1, +1]$, and for X-TFC $x \in [0, 1]$ according to Equations (3)–(5). The mapping coefficient is $c = \dfrac{x_f - x_0}{t_f - t_0}$.

With the function now expressed in terms of $\zeta$ the differential equation can now be transformed into a function of $\zeta$,

$$\left\{ c^2 \ddot{h}(x) + h(x) - \phi_1(x) h_0 - \phi_2(x) \frac{1}{c} \int_{-1}^{1} h(x) \, \mathrm{d}x \right\}^{\mathrm{T}} \zeta = -\phi_1(x) - \pi \phi_2(x).$$

This equation can then be specified for a discrete values of $x$. This discretization yields to an over-determined linear system that can be solved by least-squares.

$$\mathbb{A}\,\zeta = b \qquad \rightarrow \qquad \zeta = (\mathbb{A}^{\mathrm{T}}\mathbb{A})^{-1}\mathbb{A}^{\mathrm{T}}\,b. \tag{12}$$

In this example, using $n = 100$ and $m = 20$, the classic TFC was executed with a computational time is of $\mathcal{O}(10^{-4})$ s, the average absolute error on the discretization points is of $\mathcal{O}(10^{-16})$, the variance of the absolute error on the discretization points is of $\mathcal{O}(10^{-16})$. For X-TFC, we set the following hyper-parameters $n = 100$, $L = 100$, Gaussian activation function, input weight and bias were sampled from $\mathcal{U}[-10, +10]$ . The computational time is of $\mathcal{O}(10^{-4})$ s, the average absolute error on the discretization points is of $\mathcal{O}(10^{-16})$, the variance of the absolute error on the discretization points is of $\mathcal{O}(10^{-16})$. Table 1 reports the absolute error with respect to the analytical solutions obtained with classic TFC and X-TFC on 11 points uniformly distributed in the $[0, 1]$ range.

**Table 1.** Absolute errors on uniform test points for problem #1.

| $t/\pi$ | TFC | X-TFC |
|---|---|---|
| 0.0 | 0.0 | 0.0 |
| 0.1 | 0.0 | $2.22 \times 10^{-16}$ |
| 0.2 | 0.0 | $2.22 \times 10^{-16}$ |
| 0.3 | 0.0 | 0.0 |
| 0.4 | 0.0 | 0.0 |
| 0.5 | 0.0 | 0.0 |
| 0.6 | $4.44 \times 10^{-16}$ | $4.44 \times 10^{-16}$ |
| 0.7 | $1.11 \times 10^{-16}$ | $2.22 \times 10^{-16}$ |
| 0.8 | $6.66 \times 10^{-16}$ | 0.0 |
| 0.9 | $1.66 \times 10^{-16}$ | $2.77 \times 10^{-16}$ |
| 1.0 | $2.22 \times 10^{-16}$ | $6.66 \times 10^{-16}$ |

*3.3. Mixed Constraints*

Consider a function subject to a value-level, relative, and integral constraint,

$$f(t_1) = f_1, \qquad f(t_0) = f(t_f), \qquad \text{and} \qquad \frac{1}{t_f - t_0} \int_{t_0}^{t_f} f(\tau) \, \mathrm{d}\tau = I.$$

By using Equation (2), the constrained expression has the form,

$$f(t, g(t)) = g(t) + \phi_1(t)\,\rho_1(t, g(t)) + \phi_2(t)\,\rho_2(t, g(t)) + \phi_3(t)\,\rho_3(t, g(t))$$

where the projection functionals, $\rho_k(t, g(t))$, are given by

$$\begin{cases} \rho_1 = f_1 - g(t_1) \\ \rho_2 = g(t_f) - f(t_0) \\ \rho_3 = I - \dfrac{1}{t_f - t_0} \displaystyle\int_{t_0}^{t_f} g(\tau) \, \mathrm{d}\tau \end{cases}.$$

Given the support functions,

$$s_1(t) = 1, \qquad s_2(t) = t, \qquad \text{and} \qquad s_3(t) = t^2$$

the $\alpha_{ji}$ coefficients of the switching functions, $\phi_j(t) = \sum_{i=1}^{3} \alpha_{ji} s_i(t)$, are computed by inverting the support matrix. Specifically,

$$\begin{bmatrix} \alpha_{11} & \alpha_{12} & \alpha_{13} \\ \alpha_{21} & \alpha_{22} & \alpha_{23} \\ \alpha_{31} & \alpha_{32} & \alpha_{33} \end{bmatrix} \begin{bmatrix} s_1(t_1) & s_1(t_0) - s_1(t_f) & \frac{1}{t_f - t_0} \int_{-\pi}^{+\pi} s_1(\tau) \, d\tau \\ s_2(t_1) & s_2(t_0) - s_2(t_f) & \frac{1}{t_f - t_0} \int_{-\pi}^{+\pi} s_2(\tau) \, d\tau \\ s_3(t_1) & s_3(t_0) - s_3(t_f) & \frac{1}{t_f - t_0} \int_{-\pi}^{+\pi} s_3(\tau) \, d\tau \end{bmatrix} = \begin{bmatrix} 1 & 0 & 0 \\ 0 & 1 & 0 \\ 0 & 0 & 1 \end{bmatrix}.$$

Once the $\phi_j(t)$ are known, the constrained expression can be obtained as:

$$f(t, g(t)) = g(t) + \phi_1(t) \left( f_1 - g(t_1) \right) + \phi_2(t) \left( f(t_f) - g(t_0) \right) +$$
$$+ \phi_3(t) \left( I - \frac{1}{t_f - t_0} \int_{t_0}^{t_f} g(\tau) \, d\tau \right).$$

Problem #2

As an ultimate "stress-test" of the TFC method, a mixed-constraint case is considered where point, relative, and integral constraints are used in the solution of a differential equation, on the range $t \in [-\pi, \pi]$

$$\ddot{y} + \sin t \, \ddot{y} + (1 - t)\dot{y} + ty = f(t) \qquad \text{subject to:} \quad \begin{cases} y(t_0) = 0 \\ y(t_f) = y(t_0) \\ \displaystyle\int_{-\pi}^{+\pi} y(\tau) \, d\tau = -2\pi \end{cases}$$

where the forcing term is

$$f(t) = (t - 1) \sin^2 t + (2 + 2t - t^2 - 2 \cos t) \sin t + t(t - 1) \cos t.$$

This problem admits the analytical solution,

$$y = (1 - t) \sin t.$$

Following the procedure explained, the constrained expression for this problem is,

$$y(t, g(t)) = g(t) + \phi_1(t)(-g_0) + \phi_2(t)(-g_f) + \phi_3(t) \left( -2\pi - \int_{-\pi}^{+\pi} g(\tau) \, d\tau \right),$$

where the $\alpha_{ji}$ coefficients of the switching functions $\phi_j(t) = \sum_{i=1}^{3} \alpha_{ji} s_i(t)$ are computed by solving the following system:

$$\begin{bmatrix} \alpha_{11} & \alpha_{12} & \alpha_{13} \\ \alpha_{21} & \alpha_{22} & \alpha_{23} \\ \alpha_{31} & \alpha_{32} & \alpha_{33} \end{bmatrix} \begin{bmatrix} s_1(t_0) & s_1(t_f) & \int_{-\pi}^{+\pi} s_1(\tau) \, d\tau \\ s_2(t_0) & s_2(t_f) & \int_{-\pi}^{+\pi} s_2(\tau) \, d\tau \\ s_3(t_0) & s_3(t_f) & \int_{-\pi}^{+\pi} s_3(\tau) \, d\tau \end{bmatrix} = \begin{bmatrix} 1 & 0 & 0 \\ 0 & 1 & 0 \\ 0 & 0 & 1 \end{bmatrix}.$$

Now, expressing the free function according to $g(t) = \boldsymbol{\xi}^{\mathrm{T}} \boldsymbol{h}(x(t))$ and rearranging terms leads us to the final constrained expression,

$$y(x, \boldsymbol{\xi}) = \left\{ \boldsymbol{h}(x) - \phi_1(x)\, \boldsymbol{h}_0 - \phi_2(x)\, \boldsymbol{h}_f - \phi_3(x)\, \frac{1}{c} \int_{x_0}^{x_f} \boldsymbol{h}(x)\, \mathrm{d}x \right\}^{\mathrm{T}} \boldsymbol{\xi} - 2\pi\, \phi_3(x),$$

and by substituting this constrained expression into the differential equation (by evaluating its derivatives accordingly), we obtain a linear system that can be solved by least-squares using Equation (12).

Even in this case, the differential equation and boundary conditions must be mapped onto $x \in [x_0, x_f]$. For classic TFC $x \in [-1, +1]$, and for X-TFC $x \in [0, 1]$ according to Equation (3), where the mapping coefficient is $c = \dfrac{x_f - x_0}{t_f - t_0}$.

For classic TFC we set $n = 35$ and $m = 30$. The computational time is of $\mathcal{O}(10^{-4})$ s, the average absolute error on the discretization points is of $\mathcal{O}(10^{-15})$, the variance of the absolute error on the discretization points is of $\mathcal{O}(10^{-30})$. For X-TFC we set the following hyper-parameters $n = 40$, $L = 40$, sinusoidal activation function, input weight and bias were sampled from $\mathcal{U}[-20, +20]$. The computational time is of $\mathcal{O}(10^{-4})$ s, the average absolute error on the discretization points is of $\mathcal{O}(10^{-15})$, the variance of the absolute error on the discretization points is of $\mathcal{O}(10^{-30})$.

### 3.4. Discussions

The results from these two problems emphasize the utility and convenience of using TFC. Once the constrained expression is defined, the method to solving the differential equation does not change regardless of the use of different constraints. Moreover, this makes the solution accuracy only dependent on the problem complexity. Finally, once the solution is computed (on the discretization points), we have an analytical representation of it. That is, no further manipulations are needed (e.g., interpolation) if we want to evaluate the solutions in points that are different from the discretization points. Indeed, as shown in the results of both problems we do not lose any accuracy when evaluating the solution on test points, as it would happen with some other state-of-the-art methods such as the Finite Element Method (FEM) [34].

### 4. TFC for Linear Ordinary Integro-Differential Equation

In this section we show how TFC is applied to solve linear ordinary Integro-Differential Equations (IDEs) using all constrained expressions derived from the formalism of Equation (1).

As first test problem, the linear Fredholm integro-differential equations is considered. Comparisons of the numerical results obtained with TFC, X-TFC, and the method published in Ref. [35] are presented.

### 4.1. Problem #1

Consider the following integro-differential equation:

$$\dot{y}(x) = 1 - \frac{1}{3}x + x \int_0^1 \tau\, y(\tau)\, \mathrm{d}\tau$$

with one constraint $y(0) = 0$, for $x \in [0, 1]$. The analytical solution is $y(x) = x$. The constrained expression for this problem is simply

$$y(x, \boldsymbol{\xi}) = (\boldsymbol{h}(x) - \boldsymbol{h}_0)^{\mathrm{T}} \boldsymbol{\xi},$$

where $\boldsymbol{h}_0$ is the vector of the basis function computed at $x = 0$.

For classic TFC $n = 100$ discrete points and $m = 29$ basis functions were used. The computational time is of $\mathcal{O}(10^{-4})$ s, the average absolute error on the discretization points is of $\mathcal{O}(10^{-5})$, and the variance of the absolute error on the discretization points is

of $\mathcal{O}(10^{-9})$. For X-TFC the hyper-parameters were set to $n = 20$, $L = 20$, the activation function was sinusoidal, and the input weight and bias were sampled from $\mathcal{U}[-1, +1]$. The computational time obtained was of $\mathcal{O}(10^{-4})$ s, the average absolute error on the discretization points of $\mathcal{O}(10^{-4})$, the variance of the absolute error on the discretization points of $\mathcal{O}(10^{-8})$. The absolute errors obtained with classic TFC and X-TFC on 10 test points are reported in Figure 2 and compared with those reported by Ref. [35].

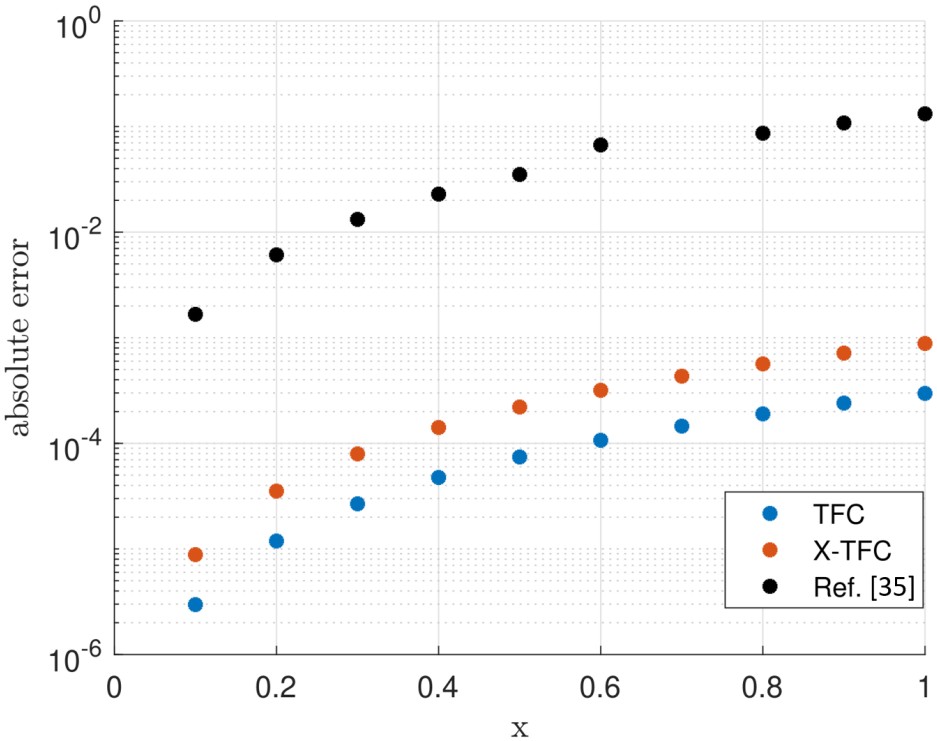

**Figure 2.** Absolute error on test points for problem #1.

*4.2. Problem #2*

The second test problem is on the following integro-differential equation:

$$\dot{y}(x) = x\,e^x + e^x - x + x \int_0^1 y(\tau)\,\mathrm{d}\tau,$$

with one constraint $y(0) = 0$, for $x \in [0, 1]$. The analytical solution is $y(x) = xe^x$. Also for this this problem the constrained expression is

$$y(x, \boldsymbol{\xi}) = (\boldsymbol{h}(x) - \boldsymbol{h}_0)^{\mathrm{T}}\boldsymbol{\xi}.$$

For classic TFC $n = 25$ discretization points and $m = 14$ basis functions were adopted. The computational time obtained is of $\mathcal{O}(10^{-4})$ s, the average absolute error on the discretization points is of $\mathcal{O}(10^{-16})$, and the variance of the absolute error on the discretization points is of $\mathcal{O}(10^{-32})$. For X-TFC the hyper-parameters were set to $n = 50$, $L = 50$, the activation function was the hyperbolic tangent, and the input weight and bias were sampled from $\mathcal{U}[-1, +1]$. The computational time is of $\mathcal{O}(10^{-4})$ s, the average absolute error on the discretization points is of $\mathcal{O}(10^{-14})$, and the variance of the absolute error on the discretization points is of $\mathcal{O}(10^{-27})$. The absolute errors on the test points obtained with classic TFC and X-TFC, are reported in Figure 3 and compared with those reported by Ref. [35].

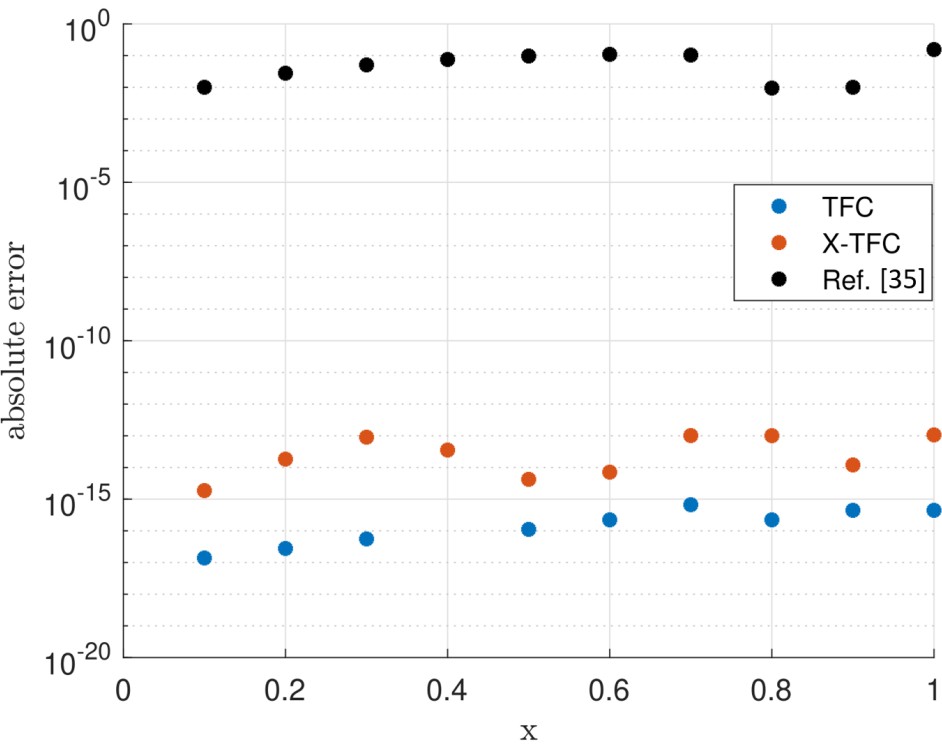

**Figure 3.** Absolute error on test points for problem #2.

*4.3. Problem #3*

The last test problem is a second-order integro-differential equation,

$$\ddot{y}(x) = e^x - x + x \int_0^1 \tau\, y(\tau)\, \mathrm{d}\tau,$$

subject to the initial value problem, $y(0) = 1$ and $\dot{y}(0) = 1$, where $x \in [0, 1]$. The analytical solution is $y(x) = e^x$. The constrained expression for this problem is:

$$y(x, \boldsymbol{\xi}) = \left(\boldsymbol{h}(x) - \boldsymbol{h}_0 - x\, \dot{\boldsymbol{h}}_0\right)^{\mathsf{T}} \boldsymbol{\xi} + x + 1.$$

To solve this problem, the classic TFC required $n = 9000$ points and $m = 1000$ basis functions. The computational time was of $\mathcal{O}(1)$ second, the average absolute error on the discretization points was of $\mathcal{O}(10^{-5})$, and the variance of the absolute error on the discretization points was of $\mathcal{O}(10^{-10})$. For X-TFC the setting was $n = 90$ and $L = 94$ hyper-parameters, hyperbolic sine activation function, and input weight and bias were sampled from $\mathcal{U}[-1, +1]$. The computational time was of $\mathcal{O}(10^{-4})$ s, the average absolute error on the discretization points of $\mathcal{O}(10^{-4})$, and the variance of the absolute error on the discretization points of $\mathcal{O}(10^{-7})$. The absolute errors on the test points obtained with classic TFC and X-TFC, are reported in Figure 4 and compared with those reported by Ref. [35].

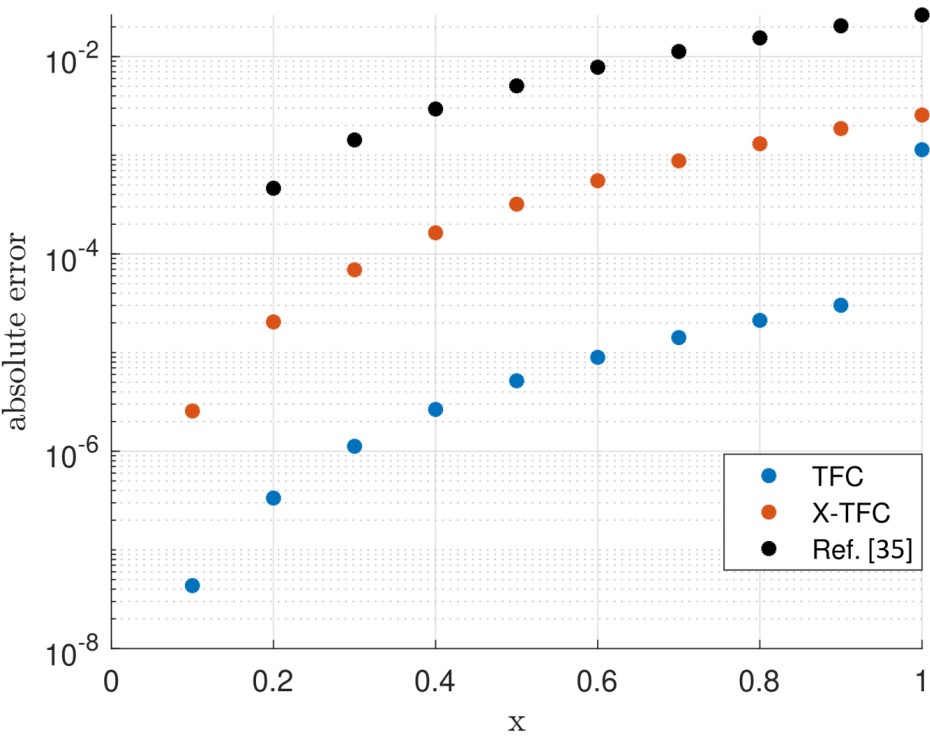

**Figure 4.** Absolute error on test points for problem #3.

### 4.4. Discussions

The discussions made for the linear ODE subjects to integral constraints are also valid for linear ordinary integro-differential equations. Nevertheless, the attentive reader will notice that there is a significant difference in terms of accuracy among problems 1 and 3 compared to problem 2. This is due to the fact that the kernels in the integral terms of the equations are different than one for problems 1 and 3 (e.g., for the both problems the kernel is $t$). This can have a negative impact on the solution accuracy as the function to be integrated can become very complex, and this can lead to numerical issues. For instance, when classic TFC is used in problems 1 and 3, where the kernel is simply $t$, the coefficients in front of the integrated polynomials become enormous as the number of basis functions (e.g., the degree of the Chebyshev polynomials) reaches above ten, causing numerical issues that impact negatively on the accuracy. Moreover, in many real world applications, such as in radiative transfer equation [17,36], the kernel can be very complex, making the analytical evaluation of the integral prohibitive.

The are several ways of mitigating these issues that are worth exploring. For instance, the integral could be evaluated with a Gauss-quadrature scheme or with a Monte Carlo method. The authors of this manuscript are investigating the possibility of evaluating the integral numerically using NNs. However, the investigation of different ways to overcome these limitations is not the focus of this work, and it will be explored by the authors in future papers.

### 5. Conclusions

This study presents an extension of the least-squares solution of linear differential equations studied in an earlier publication [5]. This paper aims to explore solutions using different boundary constraints including multi-valued, relative, and integral constraints for linear ordinary differential equations and for linear ordinary integro-differential equations. In all problems, a *constrained expression* is used to embed the constraints. This allows the integration range to be independent from the constraints location. In these expressions, the free function $g(x)$ was expressed as a linear combination of known basis functions and

unknown constant coefficients $\xi$, either using Chebyshev polynomials, or shallow NNs for the X-TFC framework. Expressing $g(x)$ as a linear combination of basis functions and constant coefficients, the coefficient vector $\xi$ remains linear and is solved by using a least-squares method. While this paper only solves linear differential equations as a test case, the proposed methodology can easily be extended to nonlinear differential equations by replacing the least-squares method with a nonlinear least-squares technique as introduced in prior research [6].

For all cases explored, the classic TFC and X-TFC methods consistently produce very fast and accurate solutions. Additionally, for all problems, the constraints are *analytically* satisfied since they are integrated into the *constrained expression*. In general, this means that the TFC methods decouple the constraints from the solution of the differential equation. Due to this, the solution range of the differential equation and, where the constraints are specified, is independent. This fact makes the TFC method a unified framework to solve differential equations with no more distinctions between the initial and boundary values problems, as well as any other constraints distribution problem.

**Author Contributions:** Conceptualization, D.M.; methodology, D.M., M.D.F., E.S. and A.D.; software, M.D.F., E.S., A.D. and D.M.; validation, M.D.F., E.S. and D.M.; formal analysis, D.M., M.D.F. and E.S.; writing—original draft preparation, D.M., M.D.F., E.S. and A.D.; writing—review and editing, D.M., E.S. and M.D.F.; supervision, R.F. and D.M. All authors have read and agreed to the published version of the manuscript.

**Funding:** This research received no external funding.

**Acknowledgments:** The authors acknowledge Hunter Johnston for the initial investigation on integral constraints using the Theory of Functional Connections.

**Conflicts of Interest:** The author declares no conflict of interest.

## Appendix A. Chebyshev and Legendre Orthogonal Polynomials

This appendix provides compact summaries of Chebyshev, $T_k(x)$, and Legendre, $L_k(x)$, orthogonal polynomials, which are defined in the $x \ß [-1, +1]$ range. These are:

*Appendix A.1. Definition*

Starting with $T_0 = L_0 = 1$ and $T_1 = L_1 = x$ these orthogonal polynomials can be conveniently defined recursively,

$$T_{k+1} = 2\,x\,T_k - T_{k-1} \qquad \text{and} \qquad L_{k+1} = \frac{2k+1}{k+1}\,x\,L_k - \frac{k}{k+1}\,L_{k-1}$$

and, specifically ($\forall k$),

$$T_k(-1) = L_k(-1) = (-1)^k \qquad \text{and} \qquad T_k(+1) = L_k(+1) = 1.$$

*Appendix A.2. Orthogonality*

The orthogonality is provided by the following integrals,

$$\int_{-1}^{+1} T_i(x)\,T_j(x)\,\frac{\mathrm{d}x}{\sqrt{1-x^2}} = \begin{cases} 0 & \text{if } i \neq j \\ \pi & \text{if } i = j = 0 \\ \pi/2 & \text{if } i = j \neq 0 \end{cases}$$

$$\int_{-1}^{+1} L_i(x)\,L_j(x)\,\mathrm{d}x = \frac{2}{2j+2}\,\delta_{ij}.$$

*Appendix A.3. Derivatives*

All derivatives of Chebyshev orthogonal polynomials can also be computed recursively. Starting from,

$$\frac{dT_0}{dz} = 0, \qquad \frac{dT_1}{dz} = 1, \qquad \text{and} \qquad \frac{d^d T_0}{dz^d} = \frac{d^d T_1}{dz^d} = 0 \quad (\forall\, d > 1),$$

the subsequent derivatives can be computed by:

$$\frac{d^d T_{k+1}}{dz^d} = 2\left( d\,\frac{d^{d-1} T_k}{dz^{d-1}} + z\,\frac{d^d T_k}{dz^d} \right) - \frac{d^d T_{k-1}}{dz^d} \quad (k \geq 1,\ \forall\, d \geq 1).$$

The derivatives of Legendre orthogonal polynomials can also be computed recursively. Starting from,

$$\frac{dL_0}{dz} = 0, \qquad \frac{dL_1}{dz} = 1, \qquad \text{and} \qquad \frac{d^d L_0}{dz^d} = \frac{d^d L_1}{dz^d} = 0 \quad (\forall\, d > 1),$$

the subsequent derivatives can be computed by,

$$\frac{d^d L_{k+1}}{dz^d} = \frac{2k+1}{k+1}\left( d\,\frac{d^{d-1} L_k}{dz^{d-1}} + z\,\frac{d^d L_k}{dz^d} \right) - \frac{k}{k+1}\,\frac{d^d L_{k-1}}{dz^d} \quad (k \geq 1,\ \forall\, d \geq 1).$$

*Appendix A.4. Integral*

- *Chebyshev indefinite.*

$$\int_{-1}^{x} T_0(z)\,dz = x + 1, \qquad \int_{-1}^{x} T_1(z)\,dz = \frac{1}{2}\left( x^2 - 1 \right),$$

$$\text{then} \qquad \rightarrow \qquad \int_{-1}^{x} T_k(z)\,dz = \frac{k\,T_{k+1}}{k^2 - 1} - \frac{x\,T_k}{k - 1}$$

- *Chebyshev full range.*

$$\int_{-1}^{+1} T_k(x)\,dx = \begin{cases} 0 & \text{for } k \text{ odd} \\[2mm] \dfrac{(-1)^k + 1}{1 - k^2} & \text{for } k \text{ even} \end{cases}$$

- *Chebyshev internal range* $(-1 \leq a < b \leq +1)$

$$\int_{a}^{b} T_k(x)\,dx = \frac{k\left[ T_{k+1}(b) - T_{k+1}(a) \right]}{k^2 - 1} - \frac{b\,T_k(b) - a\,T_k(a)}{k - 1}$$

- *Legendre indefinite.*

$$\int_{-1}^{x} L_0(z)\,dz = x + 1, \qquad \int_{-1}^{x} L_1(z)\,dz = \frac{1}{2}\left( x^2 - 1 \right),$$

$$\text{then} \qquad \rightarrow \qquad \int_{-1}^{x} L_k(z)\,dz = \frac{L_{k+1}(x) - L_{k-1}(x)}{2k + 1}$$

- *Legendre full range.*

$$\int_{-1}^{+1} L_0(x)\,dx = 2 \qquad \text{and} \qquad \int_{-1}^{+1} L_k(x)\,dx = 0, \quad \forall\, k \neq 0.$$

- *Legendre internal range* $(-1 \leq a < b \leq +1)$

$$\int_a^b L_k(x)\,\mathrm{d}x = \frac{L_{k+1}(b) - L_{k+1}(a) + L_{k-1}(a) - L_{k-1}(b)}{2k+1}.$$

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
