# Peer review of "Theory of Functional Connections Applied to Linear ODEs Subject to Integral Constraints and Linear Ordinary Integro-Differential Equations"

_mca, doi:10.3390/mca26030065_

Round 1

Reviewer 1 Report

The authors show how to solve linear ordinary differential equations and linear integro-differential equations by using a new mathematical framework to perform functional interpolation. The results are very interesting and seem to be correct. The paper is well organized. But there are some errors in typos and grammars. For examples,

(1) (77), is a set of

(2) (95), By setting the range ratio   

(3) (95), variables is linear   

(4) p.4(-7), , the derivatives    

(5) p.5(-5), By integrating    

(6) p.4(-1); p.5(+7); p.8(-5),`,’ should be replaced by `.’

(7) p.3(-14; +14); p.4(+20); p.5(-1); p.7(+12; +17; -7); p.8(+4; +7); p.8(-1); p.9(+5; +9; -1), add `.’  

(8) p.6(-9), remove `.’

(9) p.8(-4), By using  

(10) p.9(+8), remove `+’

(11) Ref [1] theory of connections: Connecting points. Mathematics 2017, 5, XXXX (paper number?

(12) Ref [3] multivariate theory of functional connections: An $n$-dimensional constraint embedding technique applied to partial differential equations.

(13) Ref [4] Least-square solution of linear differential equations. Mathematics 2017, 5, Xxxx (paper number?)

(14) Ref [5] accuracy least-squares solutions of nonlinear differential equations

(15) Ref [6] theory of functional connections applied to component constraints

(16) Ref [7] theory of functional connections:

(17) Ref [10] theory of functional connections: A new method for estimating the solutions of partial differential equations.

(18) Ref [16] Physics-informed neural networks and functional interpolation for data-driven parameters discovery of epidemiological compartmental models.

(19) Refs [20], [25] the theory of functional connections.

(20) Ref [23] Learning and Knowledge Extraction

(21) Ref [27] propagation via the theory of functional connections.

(22) Ref [28] (check [28], please).

(23) Ref [32] neural networks with functional interpolation for optimal intercept problems.

(24) Ref [33] multivariate theory of functional connections: Theory, proofs and application in partial differential equations.

(25) Ref [34] theory of connections.

(26) Ref [36] applications.

(27) Ref [40] basic problem in radiative transfer via theory of functional connections.

The authors did not read the final version before submitting the paper.

The authors have to read the final revised version before submitting the revised version.

(or see the attached report.)

Author Response

see attached file and revised manuscript
